# Influence of Alcohol and Red Meat Consumption on Life Expectancy: Results of 164 Countries from 1992 to 2013

**DOI:** 10.3390/nu12020459

**Published:** 2020-02-12

**Authors:** Chhabi Lal Ranabhat, Myung-Bae Park, Chun-Bae Kim

**Affiliations:** 1Department of Gerontology Health and Welfare, Pai Chai University, Seo-gu, Daejeon 35345, Korea; 2Policy Research Institute, Sanogaucharan 44600, Kathmandu, Nepal; 3Global Centre for Research and Development (GCRD), Kathmandu 44600, Nepal; 4Institute for Poverty Alleviation and International Development, Yonsei University, 1 Yonseidae-Gil, Wonju-City 26493, Gangwon-Do, Korea; 5Department of Preventive Medicine, Yonsei University Wonju College of Medicine, 20 Ilsan-ro Wonju-City 26426, Gandwon-do, Korea; 6Hongcheon-County Hypertension and Diabetes Registration and Education Center, 5 Sinjangdae-ro Hongcheon-Gun 25135, Gandwon-do, Korea

**Keywords:** red meat consumption, life expectancy, alcohol consumption, longitudinal ecological analysis, cancer, heart disease

## Abstract

Background: High consumption of red meat, which is carcinogenic to humans, and misuse or abuse of alcohol drinking increase premature death and shortened life expectancy. The aim of this study was to examine the association of alcohol and red meat consumption with life expectancy (LE) by analyzing data from 164 countries using an ecological approach. Design: This was a longitudinal ecological study using data from the United Nation’s (UN) Food and Agriculture Organization (FAO) for 164 countries over the period 1992–2013. In regression analysis, the relationship of alcohol and red meat consumption with LE was estimated using a pooled ordinary least squares regression model. Alcohol and red meat consumption were measured every 5 years. Results: The consumption of alcohol and red meat in high-income countries (HIC) was about 4 times (36.8–143.0 kcal/capita/day) and 5 times (11.2–51.9 kcal/capita/day) higher than that in low-income countries (LIC). Red meat and alcohol consumption had a negative estimated effect on LE in HIC (b = −1.616 *p* = < 0.001 and b = −0.615, *p* = 0.003). Alcohol consumption was negatively associated with LE for all income groups, while positive relationships were found for all estimates associated with gross national income (GNI). Conclusions: Red meat and alcohol consumption appeared to have a negative impact on LE in high-income countries (HIC) and upper-middle-income countries (UMIC), although it had no significant association with LE in low-income countries (LIC) or lower-middle-income countries (LMIC). This study suggests reviewing the policies on the gradual reduction of alcohol abuse and the high consumption of red meat, particularly HIC and UMIC.

## 1. Research in Context

### 1.1. Evidence before This Study

Many studies have shown that high and long-term consumption of red meat is associated with different cancers (such as breast, pancreas, prostate, lung, bladder, and oral cancers), cardiovascular disease, and diabetes mellitus. Genotoxic and oxidative stress factors that occur during the digestion process can destroy DNA, resulting in adenoma and malignant tumor growth. Similarly, drinking alcohol is harmful to health. The practice of alcohol abuse appears to risk for health and premature mortality due to cardiovascular disease, stroke, different types of violence, liver disease, and so on. High premature mortality reduces the life expectancy of a country. The majority of previous researches is on mortality, morbidity, and association at the individual level.

### 1.2. Added Value of This Study

This study examined the association of alcohol and red meat consumption with life expectancy at the country level. Consumption of red meat and alcohol and life expectancy were taken at the country level. Results from country-level data for 20 years with an analysis of 164 countries are added values of this study. The strength of association quantifies the level of intervention by different categories of countries. High and middle-income countries are increasing the consumption of alcohol and red meat. They are negatively associated with life expectancy.

### 1.3. Policy Implication

High- and middle- income countries are likely to revisit their policy regarding alcohol and red meat in terms of production, sales, and consumption. Moreover, guidelines on consumption in terms of quantity, time, and supplementary food.

## 2. Introduction

Global meat production has increased rapidly (5 times from 1961 to 2014) over the past 50 years. Meat production in Asia accounts forms around 40–45 percent of global meat production, of which about 2/3 is red meat (except poultry) [1]. The majority of red meat consumption is in America, Europe, and Australia. Likewise, the highest level of alcohol consumption is in Europe and Africa, which bear the heaviest burden of disease and injury attributed to alcohol [2]. A positive correlation between red meat consumption and cancer development has been established. In particular, red meat consumption has a high correlation with colorectal cancer [3,4]. According to a recent meta-analysis of 10 cohort-studies, consumption of more than 100 g of red meat per day increases the risk of colon cancer by 17% [5]. Other types of cancer, such as pancreas, prostate, lung, and bladder cancers also have positive correlations with red meat consumption [6]. Breast cancer also has a correlation with red meat consumption [7].

Red meat has been a crucial source of nutrition for human beings since their existence [8]. All necessary nutrients to the human body such as protein, vitamin B complex, heme iron, zinc, and others are abundant in red meat. In addition, n-3 polyunsaturated and conjugated linoleic fatty acids in lean tissue of the meat are known to be beneficial to our health [9,10]. Since the 1950s, the American Heart Association has recommended reducing dietary cholesterol and saturated fat intake to prevent cardiovascular disease (CVD). Current guidelines also suggest that calories from saturated fat should not exceed 7–8% of daily intake and that cholesterol intake should not exceed 300 mg per day [11,12]. Studies suggested the association of high consumption of red meat with health risks, and it typically does not cause health problems unless overeaten. Likewise, abuse and misuse of alcohol are harmful to health [13].

The International Agency for Research on Cancer (IARC), a World Health Organization affiliated organization, has categorized red meat to group 2A in October 2015 [14]. Group 2A is defined as ‘probably carcinogenic to humans’ that includes 81 factors such as human papillomavirus type 68 and biomass fuel [3]. Red meat consumption can lead to cancer development through genotoxic and oxidative stress factors that occur during the digestion process that can destroy DNA, causing adenoma and malignant tumor growth [15,16]. Cardiovascular disease (CVD), mortality, and major causes of mortality have been consistently associated with higher red meat consumption in cohort studies [17,18]. The onset of CVD and diabetes mellitus (DM) has a positive correlation with red meat consumption [19,20,21,22]. In addition, red meat consumption is related to increases in metabolic syndrome and plasma C-reactive protein concentration [23] as well as arthritis [24]. Likewise, harmful saturated fats such as low-density lipoproteins and cholesterol that are harmful to cardiovascular health are also present in red meat [25]. Moreover, studies have shown that excessive alcohol consumption increase mortality due to coronary heart disease [26] and oral cancer [27] in the USA. A study in Russia showed that the consumption of red meats over the acceptable limit would increase premature mortality by cancer [28] and increase the risk of stroke in Japan [29]. In alcoholic patients, levels of neutrophil chemokine GRO α increase, which is fatal [30], and this enzyme significantly increases in alcoholic hepatitis livers compared with normal healthy control livers [31].

Many studies have shown that the consumption of alcohol is directly proportional to cardiovascular disease and premature mortality, ultimately shortening the life expectancy in the USA [32], UK [33], Sweden [34], and other countries [35]. Increased premature death by any cause creates stagnation of life expectancy [36]. Binge and heavy drinking may cause the liver to become inflamed and in worst-case scenarios, an inflamed liver causes cirrhosis [37], impaired brain function [38], anxiety [39], and weight gain [40]. Those conditions increase premature death and disability. There is a conceptual model about how red meat and alcohol consumption increase premature mortality based on American addiction center resource 2018 [41] and the study of Saeed Mastour Alshahrani et al. in 2019 [42] (Figure 1).

To date, previous studies have found an association of red meat and alcohol consumption with health risks based on individuals [43,44,45,46,47]. However, these findings at the level of individuals have not been reported at the group or country levels nor with a time lag. Study at an individual level highlights to estimate the outcome in particular individual health. However, study at mass and country-level would also be of special importance. The life expectancy of a country reflects the overall mortality level of a population and shows the quality of life pattern that prevails across all age groups—children/adolescents, adults, and the elderly [48]. Analysis at the individual level has a narrow scope for any public health interventions. Data on a larger scale are important to develop policy in the state, country, region, and global context. In other words, global health policies such as Millennium Development Goal (MDG), Sustainable Development Goal (SDG), and so on would adopt such types of findings to be applied in a wide scenario [49]. For resource mobilization, stewardship, and the delivery mechanism through the country and global results could be extra important [50]. Moreover, the national and global level scenario is equally important for global health system governance [51]. Our analysis of red meat and alcohol consumption status could be used for policy departure regarding producing, consuming, importing, and selling red meat and alcohol within the country and outside the country considering people’s life expectancy and other national health status indicators.

Thus, the objective of this study was to assess the effect of red meat and alcohol consumption on life expectancy by analyzing data from 1992–2013 with a global sample of countries using an ecological approach.

## 3. Methods

### 3.1. Study Design

It was an ecological observational study to compare data by country and region (geography) in line with time.

### 3.2. Source of Data

Data related to food consumption were imported from the Food and Agriculture Organization (FAO) web page. Data on gross national income (GNI) and life expectancy were imported from the World Bank (WB) data center. These data included 164 countries that were United Nations (UN) member nations as of January 2016.

### 3.3. Variables

Many factors can influence life expectancy and food consumption behavior [52]. Lifestyles, particularly food and drink behaviors, are major factors to determine health and disease [53]. At the individual level, factors that can reduce life span include age, sex, occupation, education, economic status, food and drinking habits, genetic factors, and so on. At the country level, there are limited factors. Gastrointestinal (GI) cancer is directly associated with premature mortality. Alcohol, red meat, vegetable, fruits, and calorie intake is also related to GI cancer [54]. At the country level, it is difficult to identify possible factors influencing fatal diseases that shorten the life span. Moreover, including as many as countries and data for a long period was extremely difficult. By observing country-level data from FAO, we have found co-variants such as red meat, alcohol, vegetable fruits, and calories [2,4,55]. Country income was an important variable. Thus, we included national GNI as an important co-variant.

### 3.4. Measurement of Variables

For our research, life expectancy in years was the dependent/outcome variable. Red meat consumption in kilograms (kg) per capita annually was the independent variable. Similarly, vegetable, fruit, calorie intake, and alcohol consumption in kg/capita/year were independent variables. As an economic variable, GNI per capita (in USD) was also an independent variable.

Hence, the research equation was formed as:Yit = β_1_Xit + αi + Eit
Life expectancy (Y_it_) = αi + β_1_ × *ln*Red meat_it_ + β_2_ × *ln*Vegetable and Fruit_it_ + β_3_ × *ln*Calorie_it_ + β_4_ × *ln*Alcohol_it_ + β_5_ × *ln*GNI_it_ + E_it_
where αi is (I = 1,….n), the unknown intercept for each entity (n entity-specific intercepts), β_1_ to β_n_ are coefficients for independent variables (IV), Y_it_ (life expectancy where I = entity and t = time) is time interval effect, and E_it_ is error term.

### 3.5. Statistical Analysis

In order to examine concurrent trends of red meat consumption and LE, the mean values of 164 countries in our sample were mapped as quintiles. Both our dependent and independent variables were assessed by additional adjusted analysis. Pooled ordinary least square (OLS) and cross-sectional regression models were used to assess the correlation between red meat consumption and LE. A fixed-effects model and a 5-year time lag for red meat consumption were used. The natural log of each independent variable was used in all models in order to find the strength of association.

Here, pooled analysis combined time series for several cross-sections. Pooled data were characterized by having repeated observations (most frequently years) on fixed units (most frequently states and nations). This means that pooled arrays of data were ones that combined cross-sectional data on N spatial units (countries) and T time periods (22 years) to produce a data set of N × T observations. Here, the typical range of units of analysis would be 164 countries, with each unit observed over a relatively long time period, i.e., 22 years (1992–2013). Thus, it was the best model for this research.

Similarly, we used the fixed-effect model because FE explored the relationship between predictor and outcome variables within an entity (country, person, company, etc.) [56]. Each entity (country) had its own individual characteristics that may or may not influence the predictor variable (life expectancy). When using FE, we assumed that something within the individual might impact or bias the predictor or outcome variables. Thus, we needed to control for this. This was the rationale behind the assumption of the correlation between the entity’s error term and predictor variables. FE removed the effect of those time-invariant characteristics thus that we could assess the net effect of the predictor on the outcome variable. Moreover, in the FE model, those time-invariant characteristics were unique to the individual. They should not be correlated with other individual characteristics.

**Categorization of countries:** Using definitions from the World Bank International Gateway (IGW) as of January 2016, countries were categorized into the following 4 groups: Low-income countries (LIC), lower-middle-income countries (LMIC), upper-middle-income countries (UMIC), and high-income countries (HIC) [57].

## 4. Results

### 4.1. Situation of Red Meat Consumption and Life Expectancy

Low life expectancy was found in Africa, while relatively high levels of LE were seen in the following regions: North America, Western Europe, Australia, New Zealand, and Japan (Figure 2). Low amounts of red meat consumption were found in Africa, West Asia, and South Asia, while high levels of red meat consumption were found in Australia, New Zealand, North America, South America, Western Europe, and Mongolia (Figure 3). Although there were some differences by region, most countries with high LE also had high levels of red meat consumption over the last 21 years (1992–2013). The trend for alcohol consumption was the same as that for red meat consumption. The results of this study showed that North America, Europe, Russia, and Australia had high alcohol consumption (Figure 4).

### 4.2. Descriptive Analysis

LE increased in all countries over the study period regardless of income [58]. LE increased by the highest ratio (8.6 years) in LIC. Likewise, HIC had very high amounts of red meat consumption compared to other countries. Over the last 20 years, the consumption of red meat increased slightly in UMIC and LIC, while it decreased somewhat in HIC and LMIC. Vegetable and fruit consumption increased in all income groups. Vegetable and fruit consumption have been higher in UMIC compared to that in HIC since 2010. Caloric intake has increased by 200–300 kcal in an average from HIC to LIC. Alcohol consumption increased slightly in HIC, UMIC, and LMIC, but decreased marginally in LIC. GNI has gradually increased in all income groups. However, the income gap between HIC and other groups widened (Figure 5).

The variance inflation factor (VIF) between variables ranged from 1.65 to 3.40. No VIF was more than 10. Tolerance ranged from 0.30 to 0.61, indicating that multicollinearity was not a problem (Table 1).

Over the 20-year plus study period (1992–2013), LE in HIC was the highest (76.5 years), followed by that in UMIC (70.1 years), LMIC (63.9 years), and LIC (55.3 years). The difference between the LEs of HIC and LIC was 21 years. In the case of red meat consumption, it had the highest level in HIC (51.9 kg/capita/year), which was about 5 times greater than that in LIC (11.2 kg/capita/year), followed by that in UMIC (25.6 kg/capita/year), LMIC (18.6 kg/capita/year), and LIC. For vegetable and fruit consumption, HIC (206.5 kg/capita/year) showed the highest level, followed by UMIC (189.9 kg/capita/year), LMIC (120.9 kg/capita/year), and LIC (87.4 kg/capita/year). Similarly, for caloric intake, HIC (3136.0 kg/capita/year) had the highest level of consumption, followed by UMIC (2774.6 kcal/capita/day), LMIC (2480.0 kcal/capita/day), and LIC (2181.2 kcal/capita/day). HIC had the highest level of alcohol consumption (143.0 kcal/capita/day), followed by UMIC (76.0 kcal/capita/day), LMIC (36.9 kcal/capita/day), and LIC (36.8 kcal/capita/day). GNI was the highest in HIC (23,325.3 US $/capita), followed by that in UMIC (4238.3 US $/capita), LMIC (1775.1 US $/capita), and LIC (499.4 US $/capita). (Table 2).

### 4.3. Estimates from Pooled OLS and Fixed Effect Regression Models

Estimates from pooled OLS models were: b = 2.069 (*p* = 0.000) for LIC, b = 1.365 (*p* = 0.001) for LMIC, b = 0.211 (*p* = 0. 545) for UMIC, and b = 1.831 (*p* = 0.000) for HIC. Estimates from this model indicated a beneficial relationship between vegetable and fruit consumption and LE: LIC (b = 1.631, *p* = 0.000), LMIC (b = 4.109, *p* = 0.000), UMIC (b = 5.546, *p* = 0.000), and HIC (b = 1.789, *p* = 0.000). Statistically significant relationships between caloric intake and LE were found in estimates for LIC (b = 13.894, *p* = 0.000), UMIC (b = 5.258, *p* = 0.001) and HIC (b = 3.059, *p* = 0.000). Alcohol consumption was negatively associated with LE for all income groups, while beneficial relationships were found in all estimates associated with GNI (Table 3).

Estimates from our fixed-effect model differed somewhat from those of our pooled OLS specification. There was no statistically significant relationship between red meat consumption and LE in LIC (b = 0.560, *p* = 0.490), LMIC (b = −0.013, *p* = 0.977), or UMIC (b = 0.208, *p* = 0.548). However, a negative effect was found in our estimate for HIC (b = −2.039, *p* = 0.000). The only statistically significant estimate for vegetable and fruit consumption and LE indicated a positive relationship in LMIC (b = 1.766, *p* = 0.000) and HIC (b = 1.258, *p* = 0.000). Caloric intake was only positively associated with LE in LIC (b = 28.437, *p* = 0.000) and UMIC (b = 4.079, *p* = 0.003). Alcohol consumption had a negative association with LE in estimates for all income groups, while GNI was positively associated with LE for all income groups (Table 3).

Results from our model using a 5-year trend for red meat consumption revealed a statistically significant relationship with LE in LIC (b = −1.525, *p* = 0.077) or LMIC (b = −0.536, *p* = 0.353). However, red meat consumption was negatively associated with LE in UMIC (b = −1.384, *p* = 0.001) and HIC (b = −1.616, *p* = 0.000). Vegetables and fruit consumption was not statistically associated with LE for any income category, although caloric intake had positive relationships with LE in LIC (b = 35.199, *p* = 0.000), LMIC (b = 8.269, *p* = 0.001), UMIC (b = 8.818, *p* = 0.000), and HIC (b = 11.531, *p* = 0.000). Alcohol consumption had negative impacts on LE only in LIC (b = −1.562, *p* = 0.003) and HIC (b = −0.615, p = 0.003). GNI was positively associated with LE in all income groups (Table 3).

## 5. Discussion

Consumption of red meat over the acceptable level is positively associated with cancer [59], and abuse or misuse of alcohol positively increases cardiovascular disease and premature death. Principally, this fact has been established in our findings too. From a nutritional perspective, red meat is regarded as a food that should be consumed with other ingredients. It should not be completely avoided [60]. However, dietary recommendations should not be decided based on studies conducted at the individual level. For example, studies using ecological approaches to examine red meat consumption and rheumatoid arthritis have revealed a beneficial relationship [61], while studies based on analyses of individuals found no statistically significant correlation [62].

Our results indicated a negative relationship between red meat consumption and LE in UMIC and HIC. This finding was consistent with previous studies suggesting that consumption of an appropriate amount of red meat for developed countries [63,64]. However, animal source foods (ASF) such as red meat is still very important for developing countries from a nutritional point of view. Refraining from red meat consumption in these countries is not warranted [8,65,66]. Most of the African countries lie in low income and middle-income countries and in those countries such countries did not consume red meat even at the beneficiaries level, and life expectancy counted low due to high undernutrition [67,68]. The 5-year lag for red meat consumption in UMIS was negatively related to LE, although we did not find a significant correlation without the lag model. Nonetheless, in UMIC, the intake of red meat should be adequately controlled. China is classified as a UMIC. It is one of the world’s largest meat consumers [69,70]. A large Chinese cohort study has reported that an increase in the consumption of red meat is associated with increased mortality [71], similar to the result of this study. In addition, based on the recommended daily intake of red meat (71 g) in the WCRF guidelines [72], UMIC (69.8 g) and HIC (76.3 g) with similar or slightly higher red meat consumption than recommended values had negative relationships with LE that were statistically significant, while no statistically significant relationship between red meat consumption and LE was found in LIC (54.6 g) or LMIC (65.5 g) with lower levels of red meat consumption.

There are many consistent findings concerning alcohol consumption and premature mortality due to cardiovascular disease, stroke, intentional, and unintentional injury. Studies on the global burden of disease [73], retrospective population-based study of Canada, long time cross-sectional study of Romania, prospective cohort study from Russia [74], and WHO Global status report on alcohol and health 2018 [2] have revealed that alcohol consumption is reducing premature mortality and life expectancy. Such findings are similar to the results of the present study. A study by Ranabhat et al. from 194 countries showed that alcohol consumption was positively associated with adult mortality [75], coinciding with our results.

GNI showed a strong positive correlation with LE, consistent with previous studies reporting that the economic level had a positive effect on national health [76,77,78]. In our 5-year lag model, alcohol consumption was negatively correlated with LE in HIC. However, it was not correlated with LE of LIC, LMIC, or UMIC. In the pooled OLS model and lag model without time, alcohol consumption was negatively correlated with LE in all income groups. In other ecological studies, alcohol consumption also had a negative impact on health [79]. Calorie intake had a positive impact on LE overall. The average caloric intake in HIC was about 3130 kcal/day, almost 1.5 times of calorie intake in LIC at 2165 kcal/day, suggesting that HIC consumed relatively calorie-rich food compared to LIC. It seems clear that the overconsumption of calories has a negative impact on health. Nevertheless, higher caloric intake was associated with higher LE in HIC. The recommended range of caloric intake per day differs by country. However, the maximum caloric intake for an active person is 3200 kcal/day [80,81]. The average caloric intake in HIC did not appear to deviate greatly from the maximum value. According to estimates from our pooled OLS analysis, vegetable and fruit consumption had a positive relationship with LE in all income groups. However, in longitudinal analysis, vegetable and fruit consumption only had a weak relationship with LE.

Although red meat consumption has been studied as a major health risk factor, some countries with an influential meat industry are still opposed to reducing red meat consumption. Red meat is a factor that simultaneously provides benefits and detrimental to health. It seems that red meat consumption in developed countries has a negative impact on LE (1992–2013). There should be a policy discourse on the consumption of red meat and alcohol with standard amounts and time. Likewise, further research is necessary to explore the impact nationally and sub nationally.

Our study was based on ecological analyses of data from 164 countries over a 20-year period, making these results generalizable. It provides important evidence for nutritional recommendations and policy interventions. However, this study has limitations that warrant discussion. First, it was limited to information on red meat consumption from 1992 to 2011 as information from 1991 or earlier was not available in our data source. Second, our results were based on the consumption of red meat. They should not be generalized to other types of meat, such as white meat, processed meat, and other ASF. Third, other key variables, such as physical activity and smoking rates, were excluded due to the lack of data. Lastly, data on mortality in each country were insufficient in our data source to be used as an outcome measure. If specific mortality factors known to have a strong correlation with red meat such as colorectal and prostate cancer or CVD are analyzed, richer conclusions could be drawn.

## 6. Conclusions

The rationale for limiting the intake of red meat has been based mostly on studies of individuals. Our study fills a gap in the literature by providing evidence on red meat consumption and LE based on ecological analyses. Our results suggest that high consumption of red meat has a negative impact on LE in HIC and UMIC. On the contrary, red meat consumption appears to have no influence on LE in LIC or LMIC. Therefore, high- and middle- income countries are likely to revisit their policy regarding alcohol and red meat in terms of production, sales, and consumption. Moreover, guidelines on consumption in terms of quantity, time, and supplementary food.

## Figures and Tables

**Figure 1 nutrients-12-00459-f001:**
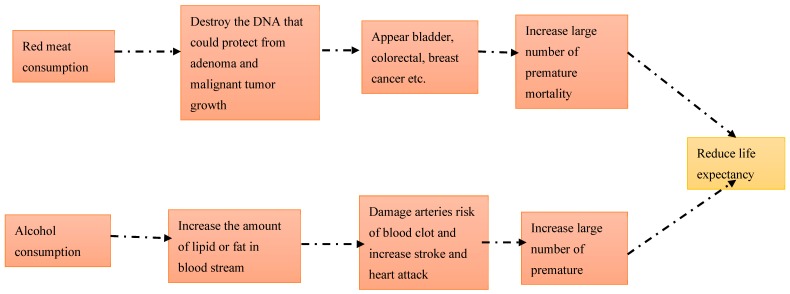
Conceptual framework on red meat and alcohol consumption associated with life expectancy.

**Figure 2 nutrients-12-00459-f002:**
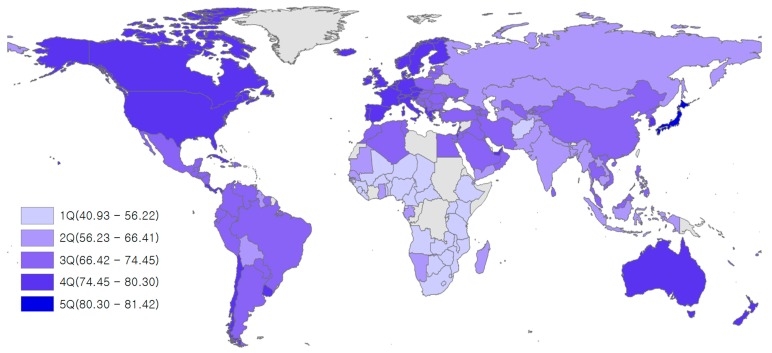
Mean life expectancy (in years) for 164 countries over the period 1992–2013.

**Figure 3 nutrients-12-00459-f003:**
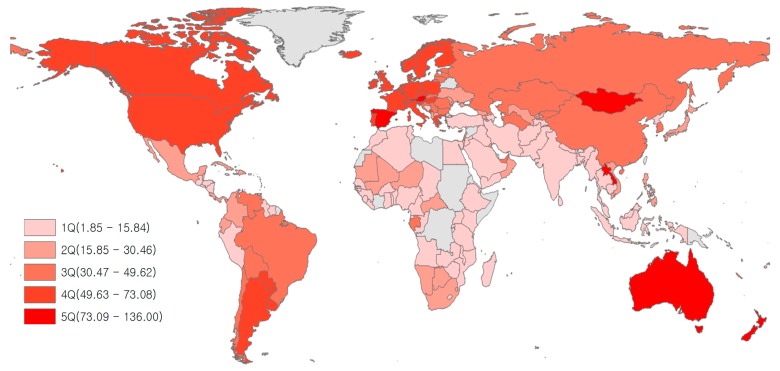
Mean red meat consumption (kg/capita/year) for 164 countries over the period 1992–2013. *Grey areas have no data.

**Figure 4 nutrients-12-00459-f004:**
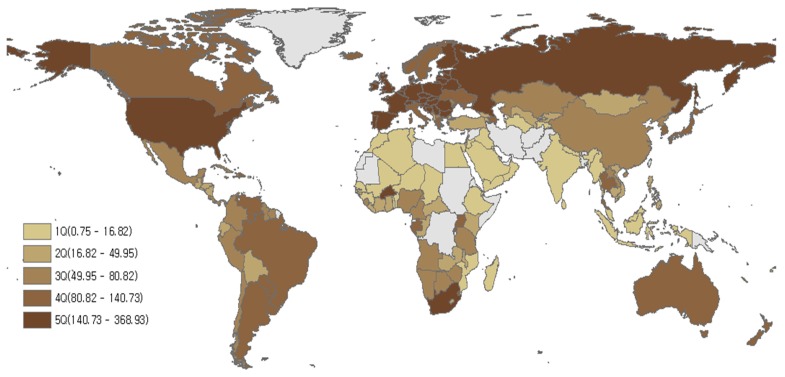
Alcohol consumption (kcal/capita/day) in 164 countries from 1992–2013. Grey areas have no data.

**Figure 5 nutrients-12-00459-f005:**
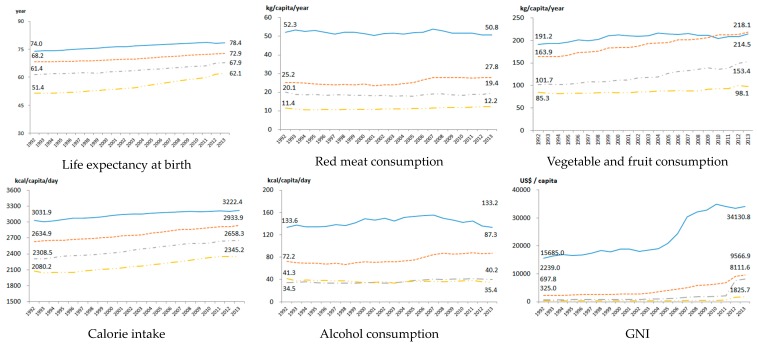
Trends of indices used to track health by the World Bank by income level over the period 1992–2013. LIC: Low-income countries; LMIC: Lower-middle-income countries; UMIC: Upper-middle-income countries; HIC: High-income countries; GNI: Gross Nation Income.

**Table 1 nutrients-12-00459-t001:** Results of multicollinearity analysis.

Variable	VIF	Tolerance
Red meat consumption	3.12	0.320
Vegetable and fruit consumption	3.06	0.327
Calorie intake	2.48	0.403
Alcohol consumption	1.89	0.528
GNI	1.64	0.611
Mean VIF	2.44	

VIF: Variance Inflation Factor; GNI: Gross Nation Income.

**Table 2 nutrients-12-00459-t002:** Means of characteristics of 164 countries by income level over the period 1992 to 2013.

	LIC (n1 = 27)	LMIC (n2 = 38)	UMIC (n3 = 47)	HIC (n4 = 52)
LE (years)	55.3	63.9	70.1	76.5
Red meat consumption (kg/capita/year)	11.2	18.6	25.6	51.9
Vegetable and fruit consumption (kg/capita/year)	87.4	120.9	189.9	206.5
Calorie intake (kcal/capita/day)	2181.2	2480.0	2774.6	3136.0
Alcohol consumption (kcal/capita/day)	36.8	36.9	76.0	143.0
GNI (US$/capita)	499.44	1775.1	4238.3	23,325.33

LE: Life expectancy at birth; GNI: Gross Nation Income; LIC: Low-income countries; LMIC: Lower-middle-income countries; UMIC: Upper-middle-income countries; HIC: High-income countries.

**Table 3 nutrients-12-00459-t003:** Adjusted estimates from regression models using pooled OLS, fixed-effects, and a 5-year lag for red meat consumption.

	Pooled OLS				Fixed-Effect Model				Lag 5 Years			
	LIC	LMIC	UMIC	HIC	LIC	LMIC	UMIC	HIC	LIC	LMIC	UMIC	HIC
Red meat consumption (kg/capita/year)	2.069 (0.000)	1.365 (0.001)	0.211 (0.545)	1.831 (0.000)	−0.560 (0.490)	−0.013 (0.977)	0.208 (0.548)	−2.039 (0.000)	−1.525 (0.077)	−0.536 (0.353)	−1.384 (0.001)	−1.616 (0.000)
Vegetable and fruit consumption (kg/capita/year)	1.63 (0.000)	4.109 (0.000)	5.546 (0.000)	1.789 (0.000)	0.914 (0.231)	1.766 (0.000)	−0.094 (0.752)	1.258 (0.000)	−0.120 (0.881)	0.245 (0.622)	−0.312 (0.396)	0.519 (0.878)
Calories intake (kcal/capita/day)	13.894 (0.000)	−0.839 (0.725)	5.258 (0.001)	3.059 (0.000)	28.437 (0.000)	2.126 (0.159)	4.079 (0.003)	3.324 (0.152)	35.199 (0.000)	8.269 (0.001)	8.818 (0.000)	11.531 (0.000)
Alcohol consumption (kcal/capita/day)	−1.124 (0.000)	−1.691 (0.000)	−0.594 (0.001)	−0.240 (0.017)	−1.909 (0.000)	−0.291 (0.217)	−0.259 (0.127)	−0.216 (0.272)	−1.562 (0.003)	0.368 (0.190)	−0.299 (0.226)	−0.615 (0.003)
GNI (US$/capita)	3.983 (0.000)	3.580 (0.000)	2.236 (0.000)	2.644 (0.000)	4.355 (0.000)	2.058 (0.000)	2.103 (0.000)	2.397 (0.000)	1.798 (0.000)	1.184 (0.000)	1.439 (0.000)	2.079 (0.000)
Cons	−84.592 (0.000)	16.455 (0.089)	−16.042 (0.122)	10.925 (0.061)	−187.375 (0.000)	25.758 (0.000)	21.847 (0.023)	−60.705 (0.000)	−221.033 (0.000)	−10.657 (0.527)	−4.941 (0.700)	−28.225 (0.000)
Adj. R-squared	0.325	0.296	0.411	0.586	0.591	0.395	0.428	0.420	0.486	0.248	0.291	0.495
Observation	551	794	934	992	551	794	934	992	434	622	723	768
Number of countries	27	38	47	52	27	38	47	52	26	38	47	52

LIC: Low-income countries; LMIC: Lower middle-income countries; UMIC: Upper-middle-income countries; HIC: High-income countries, GNI: Gross Nation Income; OLS: Ordinary Least Square.

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
