# Peer review of "Influence of Alcohol and Red Meat Consumption on Life Expectancy: Results of 164 Countries from 1992 to 2013"

_nutrients, 2020, doi:10.3390/nu12020459_

Round 1
Reviewer 1 Report
Influence of alcohol and red meat consumption on life expectancy
Overall I thought the paper has merit and a good idea to use extant data to determine if there is a correlation in population data. Throughout the paper there are blanketed statements and assertions of things like “no level of alcohol is safe for health”; “immediate regulation of red meat”; “increased social conflict” etc. I think that these should be removed as a scientific paper is not a platform for opinion but a platform to present data and unbiased analysis.
In the section on Research in context there are no references at all. It was a bit confusing to me since I had just read the abstract. While I understand the need for these sections could the not be housed in the introduction with appropriate references? You have a statement about immediate regulation but remember not all counties are able to regulate what their people consume. In democratic countries people have the freedom to choose how much and what they eat.
Introduction section
Line 81: did you mean for it to be n_3 or n-3?
Line 84: need to spell out CVD you do on the next page but this is the first time we see it
Line 86: you use the words known association this is definitive words most research highly suggests or highly correlated but typically not known association.
Line 91: the word (it) = red meat or group 2A
Line 102 chemokine GRO a = why is this important to know or to study
Line 105: opinion needs to be removed for alcohol increases social conflict.
Line 111: opinion again no level of alcohol consumption improves health. There are papers out in the literature that show that modest amounts of red wine can help with cardiovascular care
Line 113: on the reference 2018 do you want a space between the words and the ref.
If the figures are not yours you need to make sure you have references for them
Line 119: you only have 2 references but it seems like there should be more in this category
Lines 122-124: you are making assertions again that are your opinion
Lines 125-130: difficult to follow consider revising
Selection of variables – do you need this section?
Measurements of variables
Line 173: the formula- is this something you created or something that you are referencing. When I looked it up it showed as one to use for public policy for performance explanations
Results section
Line 215 and 217: Africa has low life expectancy and use of low red meat but your conclusion is those that have low life expectancy are those that eat a lot of red meat
Figure 2-5 you need references for these if you did not create them
Line 297: if the b values are negative would they be considered protective? And those that are highly positive show negative impact.
Discussion
Line 309: need reference for first sentence
Author Response
Overall, I thought the paper has merit and a good idea to use extant data to determine if there is a correlation in population data. Throughout the paper there are blanketed statements and assertions of things like “no level of alcohol is safe for health”; “immediate regulation of red meat”; “increased social conflict” etc. I think that these should be removed as a scientific paper is not a platform for opinion but a platform to present data and unbiased analysis.
Response: We greatly honor reviewer for the careful examination of our manuscript and evaluating as high merit paper that we normally expected from authors. According to your review report, our manuscript is fundamentally correct, and we need to revise some way of presentation. We have addressed the issues you raised and as researcher we are open to receive feedback and you can suggest the best approach of presentation if you feel uncomfortable with some writing. Indirectly, the published materials are the property of reviewer too.
We agree that there are some sentences as you mentioned having direct and blanketed writing and we modified those sentences and now presented with unbiased and safe way.
In the section on Research in context there are no references at all. It was a bit confusing to me since I had just read the abstract. While I understand the need for these sections could the not be housed in the introduction with appropriate references? You have a statement about immediate regulation but remember not all counties are able to regulate what their people consume. In democratic countries people have the freedom to choose how much and what they eat.
Response: Now a days there is a practice (As a researcher of global burden of disease and our study nature is in line with it) to write “Research in context” to clearly mention the novelty and basic characteristics of research outcome. In this section, there is no practice to cite any reference because those data and contents need to put in article content. Here are some examples:
https://www.thelancet.com/action/showPdf?pii=S2352-3018%2819%2930196-1 https://www.thelancet.com/action/showPdf?pii=S0140-6736%2819%2930841-4 https://www.thelancet.com/action/showPdf?pii=S0140-6736%2816%2931012-1During the publication such section can be insert as a box around introduction section, but journal format does not allow, we can remove.
As you suggest the sentences like “immediate regulation ….we have revised with fine tune.
Introduction section
Line 81: did you mean for it to be n_3 or n-3?
Response: It is n-3, we corrected.
Line 84: need to spell out CVD you do on the next page, but this is the first time we see it
Response: We have spelled out cardiovascular diseases (CVD) in this line.
Line 86: you use the words known association this is definitive words most research highly suggests or highly correlated but typically not known association.
Response: we have revised this term highly correlated as you suggested.
Line 91: the word (it) = red meat or group 2A
Response: This reflects group 2A and now we joined the sentence “ that includes 81 factors…
Line 102 chemokine GRO alpha (a) = why is this important to know or to study
Response: We have clarified the importance adding another sentence.
Line 105: opinion needs to be removed for alcohol increases social conflict.
Response: We removed this sentence.
Line 111: opinion again no level of alcohol consumption improves health. There are papers out in the literature that show that modest amounts of red wine can help with cardiovascular care
Response: We have very carefully modified the sentences with reference and removed this blanketed sentence.
Line 113: on the reference 2018 do you want a space between the words and the ref.
Response: Yes, you are right. We made space after this reference.
If the figures are not yours you need to make sure you have references for them
Response: Thanks for your suggestions and all figures we created ourselves.
Line 119: you only have 2 references but it seems like there should be more in this category
Response: We have added more references.
Lines 122-124: you are making assertions again that are your opinion
Response: The writing was like opinion and now we have modified the sentence.
Lines 125-130: difficult to follow consider revising
Response: We revised those sentences and made in smooth tone.
Selection of variables – do you need this section?
Response: We removed this section.
Measurements of variables
Line 173: the formula- is this something you created or something that you are referencing. When I looked it up it showed as one to use for public policy for performance explanations
Response: The formula we used to show the association between dependent and independent variable that we used and to show the level of association. We can use for public policy explanation too but here just variables association and level of association. In other words, we created research hypothesis based on level of association.
Results section
Line 215 and 217: Africa has low life expectancy and use of low red meat, but your conclusion is those that have low life expectancy are those that eat a lot of red meat.
Response: Actually, this is not well interpretation of our findings. Red meat up to 300 mg per day is beneficial and need even as you mentioned, limited amount of red wine would be beneficial. So, in Africa, there is no sufficient protein including meat and is may be one of the factors for under-nutrition. We have clarified such situation in discussion with reference.
Figure 2-5 you need references for these if you did not create them
Response: The applied figures in this manuscript we created ourselves, however those figures we made from data and some references. We did not completely copy the from other references.
Line 297: if the b values are negative would they be considered protective? And those that are highly positive show negative impact.
Response: It is quite theoretical question. In regression, the interpretation of beta coefficient is: An one unit increase in your regressor x decreases the estimated mean of your dependent variable y by Beta units. On other words, there is an inverse relationship, similar to that of a negative correlation. So, we can not say negative association is protective. This interpretation is useful on odds ratio not for beta coefficient. Such beta coefficient also link with p value to describe statistical significant.
Discussion
Line 309: need reference for first sentence
Response: We have added the reference.
We keenly observed your suggestions and comments, verified latest research papers and all agreed with final manuscript. If there are some errors we will minimize till author's proof. Moreover, we respect your suggestion (In some writing) to make more advance our manuscript.
Reviewer 2 Report
Line 1
Capitalize A R M C L E C in title
Line 18-19 awkward English and passive tense, consider reworking 1st sentence
Line 39-40 appears to be a subjective conclusion, one could rather posit that red meat in developed nations higher in omega 6 fatty acids leading to more inflammation as opposed to LE countries in which the animals are grass fed and pasture raised, making them higher in omega 3s and less inflammatory and less dangerous for consumption.
Line 53 Research has been done at the individual level…
Line 54 unclear
Line 69 delete rd as it is redundat, 2/3 is fine
Line 100 “Study 100 in Russia”, should be “A study in Russia”
Line 101, excessive use does not make sense, one could say excessive consumption
116, Appear wrong tense, should be appearance. Also consider reworking diagram, bottom part is cut off
156 gastrointestinal is one word
310 citation for fact?
317 is a difficult conclusion to make as it is still as subjective statement, if this were the case etoh and tobacco would be better regulated too given the studies.
348 round to nearest whole number
360 use the word health once, redundant
Author Response
Dear Reviewer
We really appreciated your valuable time to review our manuscript and our authors team acknowledged your academic contribution. We have revised the typo errors as you mentioned and other mistake we found ourselves. If you have some suggestive comments to make advance the manuscript, we really appreciate till final proof reading.
Response point by point
Line 1
Capitalize A R M C L E C in title
Response: We updated.
Line 18-19 awkward English and passive tense, consider reworking 1st sentence
Response: We updated the language.
Line 39-40 appears to be a subjective conclusion, one could rather posit that red meat in developed nations higher in omega 6 fatty acids leading to more inflammation as opposed to LE countries in which the animals are grass fed and pasture raised, making them higher in omega 3s and less inflammatory and less dangerous for consumption.
Response: we have modified the sentences based on evidence and soft flavor.
Line 53 Research has been done at the individual level…
Line 54 unclear
Response: we revised in simple sentences
Line 69 delete rd as it is redundat, 2/3 is fine
Response: We changed in 2/3.
Line 100 “Study 100 in Russia”, should be “A study in Russia”
Response: We corrected.
Line 101, excessive use does not make sense, one could say excessive consumption
Response: We corrected as excessive consumption.
116, Appear wrong tense, should be appearance. Also consider reworking diagram, bottom part is cut off
Response: We lowered the caption part balanced with diagram.
156 gastrointestinal is one word
Response: We joined those words.
310 citation for fact?
Response: We added the citation.
317 is a difficult conclusion to make as it is still as subjective statement, if this were the case etoh and tobacco would be better regulated too given the studies.
Response: We have modified the sentence with non-subjective tone.
348 round to nearest whole number
Response: We have revised as “about 3130 kcal/day
360 use the word health once, redundant
Response: We removed the double word - health
We revised the manuscript as per your comments and some minor errors we corrected and added more updated reference. Your suggestive comments in our writing (If some), we will update it before publication.
Round 2
Reviewer 2 Report
Try not to say things in longer than necessary tones, for example in line 20 unhealthy alcohol drinking can be just called "alcohol abuse or misuse"
Line 52 Unhealthy consumption of alcohol appears to increase the risk of heart and premature...
Line 66 gradual reduction in alcohol consumption, wording in red is long and difficult to understand
Line 89 health problems, s missing.
Line 90 Alcohol consumption within normal limits would not harm health...red sentence does not make sense
Line 313 does not make sense
consider consumption of red meats over the acceptable limit is associated with cancer...
323 consumption of an appropriate amount of
369 red meat is both beneficial and detrimental to health
395 incomplete sentence
Author Response
Try not to say things in longer than necessary tones, for example in line 20 unhealthy alcohol drinking can be just called "alcohol abuse or misuse"
Dear Reviewer,
We are extra grateful for your sincere observation to our manuscript. In starting version, we got the comments of subjective and directive tone in writing particularly in abstract, introduction and discussion. In Second version, we became more sincere and presented safely. Moreover, we have presented the contents in short and simple writing. The manuscript has been edited by native speaker (Prof. Margret Storey). If you point out some insufficiency of writing, please mention by PPL (Page, paragraph and line). Some space, preposition and line would be different by computer, window and version. It will be corrected before publication, but we are aware that single error needs to be corrected before publication.
Thank you, we revised it. Previously, we wanted to present safely, and some literature presented as unhealthy drinking however, it seems quite controversy.
Line 52 Unhealthy consumption of alcohol appears to increase the risk of heart and premature...
Response: Same as above, we revised as abuse of alcohol.
Line 66 gradual reduction in alcohol consumption, wording in red is long and difficult to understand
Response: We revised the sentence in a simple way.
Line 89 health problems, s missing.
Response: We added.
Line 90 Alcohol consumption within normal limits would not harm health...red sentence does not make sense
Response: We revised the sentence adhering previous sentence.
Line 313 does not make sense
consider consumption of red meats over the acceptable limit is associated with cancer...
Response: We revised as you suggested red meat over the acceptable limit and abuse or misuse of alcohol …
323 consumption of an appropriate amount of
Response: we revised as “an appropriate amount of…
369 red meat is both beneficial and detrimental to health
Response: We revised as you suggested.
395 incomplete sentence
Response: We modified the sentence in a simple way.
Again thank you for your valuable time.